# Path Loss Model for 3.5 GHz and 5.6 GHz Bands in Cascaded Tunnel Environments

**DOI:** 10.3390/s22124524

**Published:** 2022-06-15

**Authors:** Jingyuan Qian, Yating Wu, Asad Saleem, Guoxin Zheng

**Affiliations:** 1Key Laboratory of Specialty Fiber Optics and Optical Access Networks, Joint International Research Laboratory of Specialty Fiber Optics and Advanced Communication, Shanghai University, Shanghai 200444, China; jingyuan@shu.edu.cn (J.Q.); ytwu@shu.edu.cn (Y.W.); 2Shenzhen Key Laboratory of Antennas and Propagation, College of Electronics and Information Engineering, Shenzhen University, Shenzhen 518060, China; asadalvi64@szu.edu.cn

**Keywords:** ray tracing, extra loss coefficient, shooting and bouncing ray method, subway tunnel, waveguide effect

## Abstract

An important and typical scenario of radio propagation in a railway or subway tunnel environment is the cascaded straight and curved tunnel. In this paper, we propose a joint path loss model for cascaded tunnels at 3.5 GHz and 5.6 GHz frequency bands. By combining the waveguide mode theory and the method of shooting and bouncing ray (SBR), it is found that the curvature of tunnels introduces an extra loss in the far-field region, which can be modeled as a linear function of the propagation distance of the signal in the curved tunnel. The channel of the cascaded straight and curved tunnel is thus characterized using the extra loss coefficient (ELC). Based on the ray-tracing (RT) method, an empirical formula between ELC and the radius of the curvature is provided for 3.5 GHz and 5.6 GHz, respectively. Finally, the accuracy of the proposed model is verified by measurement and simulation results. It is shown that the proposed model can predict path loss in cascaded tunnels with desirable accuracy and low complexity.

## 1. Introduction

In recent years, wireless communication has become ubiquitous in areas such as tunnels, underground stations, crossing bridges, etc., to provide high-reliability and low-latency data transmission for security and train control [1] and to offer various communication or entertainment services to passengers [2]. The environment in a subway tunnel is much more complex than free space, necessitating a thorough analysis of signal transmission [3]. Therefore, research on the propagation attributes of the signals and channel models in a tunnel environment is essential for designing wireless communication systems and transmission technologies [4,5].

Channel modeling approaches in tunnel environments can be categorized into statistical methods and deterministic methods [6,7]. Classical statistical models, such as the close-in (CI) model [8] and floating intercept (FI) model [9], have low complexity and high computing efficiency, but with limited accuracy [10]. On the other hand, deterministic methods include the waveguide approach, numerical methods for solving Maxwell equations, and the RT method [11]. The waveguide approach considers the tunnel as an ideal super waveguide [12], and radio propagation in tunnels is hence regarded as multimode propagation in waveguides [13]. As idealized approximations are involved, it is difficult to match well with the real channel in complex tunnel environments [14]. Among the numerical methods for solving Maxwell equations, Finite-Difference Time-Domain (FDTD) [15] and Vector Parabolic Equation (VPE) [16] are the methods most widely used to improve the accuracy, with the cost of high complexity [17]. Finally, RT methods are based on geometric optics, where the received signal can be expressed as the sum of all rays arriving at the receiver through line-of-sight (LOS) and reflection paths [18]. The accuracy of the RT methods is reliable when three-dimensional (3D) model of the tunnels and RT parameters are set properly [19], but it requires large computing memory and additional 3D modeling for tunnel environments [20].

As listed in Table 1, considerable efforts have been devoted to improving the path loss models for different tunnel types during the past few decades [21,22,23,24,25,26,27,28,29,30,31,32,33]. For straight tunnels, the authors of [21] conducted a measurement campaign in subway tunnels at 900/2400 MHz and established a FI model by fitting the measurement results. While based on waveguide theory, a multimode model is proposed in [22], which gives an expression for the path loss at any frequency and position in tunnels. In [23], the accuracy of the SBR method for RT modeling is improved via per-ray cone angle calculation. In [24,25], surrogate models of arched tunnels are extracted in the form of rectangular waveguides by combining waveguide models and computationally intensive VPE methods. However, the validity of the surrogate models is affected by transmitter and receiver positions, especially in the regions closer to the boundary walls.

For curved tunnels, an improved CI model is proposed by introducing a breakpoint distance based on the measurement data in railway communications, which has been shown to have better accuracy but less stability than the traditional CI model [26]. In [27], channel measurements are conducted in a realistic curved-tunnel environment including the presence and absence of human bodies in terms of LOS and NLOS scenarios. An improved FI path-loss model is proposed to describe the transition between the enhanced and degraded waveguiding mechanisms in curved tunnels with the calculated break-point in [28]. Combined with waveguide theory and the RT method, [29] uses geometrical optics rules to model the main effects of the tunnel curvature and proposes a heuristic model, which can be useful for preliminary path-loss estimation. More recently, the back propagation neural network is incorporated in the model to predict the path loss in the curved tunnels [30].

For cascaded tunnels, [31] measures five frequency bands (i.e., 2.6 GHz, 3.5 GHz, 5.6 GHz, 10 GHz, and 28 GHz) in a subway tunnel composed of straight and curved sections, and a typical CI model is employed to fit the measurement results. In [32], a high-precision 3D model of a measurement tunnel is reconstructed, and the path loss is predicted using the RT method based on the 3D models. To improve the feasibility of the FDTD method, a segmented FDTD method is proposed to model radio-wave propagation in tunnels, which breaks up the computational space into segments. However, it still needs to solve sets of simultaneous equations [33].

In summary, most research on path loss model for tunnels in the literature mainly deals with a single straight or single curved tunnel, and currently there are only a few works that focus on cascaded tunnels. Meanwhile, it is worth noting that the existing schemes suffer from problems of high computational complexity, extra cost such as high-precision 3D tunnel models, excessive memory, and CPU resources.

Regarding this issue, this paper studies the propagation characteristics of radio wave in cascaded tunnels and presents a simplified path-loss model for cascaded tunnels. The extra loss in the far-field region due to the curvature is modeled as a linear function of the propagation distance of the signal in the curved tunnel. Compared with typical RT methods relying on high-precision 3D tunnel models, the proposed model can predict path loss in cascaded tunnels with comparable accuracy but significantly lower complexity.

The rest of this paper is summarized as follows. Section 2 proposes the simplified path-loss model for cascaded tunnels based on waveguide and SBR. Section 3 presents the channel measurement at 3.5 GHz and 5.6 GHz bands in a real subway tunnel environment and calibrates the material parameters of RT simulation. In Section 4, the empirical formula between the ELC and the radius of curvature is provided. Section 5 verifies the accuracy of the proposed model. Finally, the conclusion is given in Section 6.

## 2. Path Loss Model for Cascaded Tunnels in Far-Field Region of Propagation

As shown in Figure 1, the proposed research aims at cascaded tunnels composed of straight and curved tunnels. According to the proposed model, the transmitting antenna (T_x_) is placed in the straight tunnel and the distance between T_x_ and the curved tunnel is *d*, while the receiving antenna (R_x_) is placed in the curved tunnel with the radius of curvature (*R*), and the distance of R_x_ from the straight tunnel is *d*′.

### 2.1. Calculation of Break Point

Based on the waveguide model, radio propagation in tunnels can be modeled in the same way as the propagation of radio waves in a rectangular waveguide [34]. In the nearfield region, the electromagnetic field consists of multiple modes which interact and produce wide and rapid variations. In the far-field region, the fundamental mode determines the electromagnetic field [35]. Therefore, the break point (BP) is considered to divide the propagation region into a near-field region and a far-field region. This divide is determined by signal wavelength and the tunnel’s dimensions [36].
(1)$$\mathrm{BP}=\max\left(\frac{w^{2}}{\lambda },\frac{h^{2}}{\lambda }\right)$$
where *w* and *h* represent the width and height of the rectangular tunnel, and *λ* is the signal wavelength. At present, most subway tunnels are arch structured, and an arched tunnel can be equivalent to a rectangular tunnel [37].
(2)$$w=\sqrt{\frac{2s}{1+\cos\theta }[\left(\pi -\theta \right)r_{a}^{2}{+h}_{a}^{2}\tan\theta ]}$$
(3)$$\;h\;=\sqrt{\frac{s(1+\cos\theta )}{2}[\left(\pi \;-\;\theta \right)r_{a}^{2\;}{+h}_{a}^{2}\tan\theta ]}$$
where *r_a_* represents the radius of the arched tunnel, *h_a_* represents the distance from the center of the tunnel to the bottom of the tunnel, *s* and *θ* are the cross-sectional area ratio of a rectangular to a circular area, which can be calculated by the following expression.
(4)$$s=\sqrt[{3}]{{\left(x_{1}^{0}\right)}^{4}/16\pi }$$
(5)$$\theta =\arccos\left(\frac{h_{a}}{r_{a}}\right)$$
where $x_{1}^{0}$ is the first zero of the Bessel function.

### 2.2. The Joint Channel Model Based on Waveguide and SBR

Basically, SBR is used to calculate attenuation by considering the number of reflections and reflection coefficients in the tunnel. For the ray with initial power *P*_0_, the power *P* after propagating at a certain distance *z* can be expressed as [38]:(6)$$\frac{P_{0}}{P}=\frac{1}{{\left|\Gamma_{1}\right|}^{{2N}_{1}z}\cdot {\left|\Gamma_{2}\right|}^{{2N}_{2}z}}$$
where *N*_1_ and *N*_2_ represent the number of reflections per meter in the vertical and horizontal walls, and *Г*_1_ and *Г*_2_ represent the reflection coefficients in the vertical and horizontal walls. Since the tunnels have curvature along their vertical sidewalls, this study only focuses on the path-loss difference caused by the curvature of the vertical tunnel sidewalls. For vertical polarization, *Г*_1_ can be expressed as [39]:(7)$$\Gamma_{1}=\frac{\cos\;\phi_{1}-\sqrt{\cos^{2}\phi_{1}{+\varepsilon }_{r}^{*}\;-\;1}}{\cos\;\phi_{1}+\sqrt{\cos^{2}\phi_{1}{+\varepsilon }_{r}^{*}\;-\;1}}$$
where $\varepsilon_{r}^{*}$ represents the complex permittivity of the wall, and *ϕ*_1_ represents the incident angle of the ray with the vertical wall. In a straight tunnel, *ϕ*_1_ and *N*_1_ are defined as [30]:(8)$$\phi_{1}=\frac{\pi }{2}-\frac{m\lambda }{2w},\;\;\;\;m=1,\;2,\;3,\;\ldots$$
(9)$${\;N}_{1}\;\approx \;\frac{m\lambda }{{2w}^{2}},\;m=1,\;2,\;3,\;\ldots$$
where *m* represents different modes in the near-field region. Because high-order modes fade rapidly after BP, it is assumed that only the basic mode (*m* = 1) exists in the far-field region.

In the curved tunnel as shown in Figure 2, *y* represents the distance between two reflection points (*P*_1_ and *P*_2_) in the tunnel with the radius of curvature (*R*), and *ϕ*_1_ can be defined as:(10)$$\phi_{1}{=\cos}^{-1}\left(\frac{y}{2R}\right)$$

In comparison with the incident angle in the straight tunnel (*ϕ_s_*), as shown in Figure 3, the decrease in the incident angle in the curvature (*ϕ_c_*) leads to an increase in reflection times and a decrease in the reflection coefficient, which generates EL in the curved tunnel, as shown in Figure 4. The EL can be expressed as:(11)$$\;\mathrm{EL}=10\;log\left(\frac{P_{straight}}{P_{curve}}\right)\left[\mathrm{dB}\right]$$

According to (6), the EL can be rewritten as a function of the propagation distance of curvature (*z*).
(12)EL(z)=ELC·z[dB]

In (12), the ELC represents the extra loss coefficient in the curved tunnel, and its expression is as follows:(13)$$\mathrm{ELC}=20\;log\left(\frac{{\left|\Gamma_{1s}\right|}^{N_{1s}}}{{\left|\Gamma_{1c}\right|}^{N_{1c}}}\right)\;\;$$
where $\Gamma_{1s}$ and $\Gamma_{1c}$ represent the reflection coefficients along the vertical walls of the straight and curved tunnels, respectively, and $N_{1s}$ and $N_{1c}$ represent the number of reflections per meter along the vertical walls of the straight and curved tunnels, respectively.

It is assumed that the fundamental mode is reflected *M* times on the vertical wall of the curved tunnel with a length of *d’*, and *ϕ_i_* is the incident angle of the *i*-th reflection. According to (7), the average ELC can also be expressed as:(14)$${\mathrm{ELC}=20N}_{1s}\;log\left|\Gamma_{1s}\right|+{20N}_{1c}\left(log\left|\varepsilon_{r}^{*}-1\right|-2\rho \right)$$
(15)$${\;N}_{1c}=\frac{d\prime }{M}$$
(16)$$\rho =\frac{\sum_{i=1}^{M}log\left(\cos\phi_{i}-\sqrt{\cos^{2}\phi_{i}{+\varepsilon }_{r}^{*}-1}\right)}{M}$$

According to (10), the variable ρ can be rewritten as a function of the radius of curvature.
(17)$$\rho =\frac{\sum_{i=1}^{M}log\left(\frac{y_{i}}{2R}-\sqrt{{\left(\frac{y_{i}}{2R}\right)}^{2}{+\varepsilon }_{r}^{*}-1}\right)}{M}$$

To ensure the safety of high-speed traveling, the radius *R* of the curved tunnel is usually required to be at least greater than 300 m. Therefore, the variable ρ can be simplified by the Maclaurin formula.
(18)$$\rho \;\approx \;\frac{1}{ln10}\left(ln\left(-\sqrt{\varepsilon_{r}^{*}-1}\right)-\;\frac{1}{\sqrt{\varepsilon_{r}^{*}-1}}\cdot \frac{\sum_{i=1}^{M}y_{i}}{2MR}\right)$$

According to (14) and (18), the ELC can be rewritten as power function of *R*.
(19)$$\mathrm{ELC}=a+b\;\times \;\;R^{-1}$$
(20)$$\;a\;{=20N}_{1s}{log\;|\;\Gamma }_{1s}{|\;+\;20N}_{1c}{(log\;|\varepsilon }_{r}^{*}-1\;|\;-\;\frac{2ln\left(-\sqrt{\varepsilon_{r}^{*}-1}\right)}{ln10})$$
(21)$$\;b=\frac{{20N}_{1c}\;\times \;\sum_{i=1}^{M}y_{i}}{ln10\;\times \;M\;\times \sqrt{\varepsilon_{r}^{*}-1}}$$

High-order modes become significant when R_x_ is located in the near-field region, complicating the propagation phenomena and making EL difficult to determine. However, when R_x_ is in the far-field region, the propagation is mainly dominated by the fundamental mode. In addition, the EL is directly proportional to the propagation distance along the curvature in (12), and there is power function relation between ELC and R in (19). Therefore, this paper proposes a path-loss model for cascaded tunnels in the far-field region.
(22)$${PL}_{cascade}{(d+d’)=PL}_{straight}(d+d\prime )+\mathrm{ELC}(R)\;\times \;d\prime \;\left[\mathrm{dB}\right]$$
(23)d ≥ BP
where *d* represents the distance from T_x_ to the curved tunnel, as shown in Figure 1, and *d*′ represents the distance from R_x_ to the straight tunnel. ${PL}_{cascade}$ and ${PL}_{straight}$ represent the path loss of the cascaded tunnel and straight tunnel, respectively. To ensure that the curvature is completely in the far-field region of propagation, *d* must satisfy (23).

## 3. Channel Measurement and RT Parameters Calibration in a Subway Tunnel

### 3.1. Measurement Environment and Configuration

As shown in Figure 5, the measurement campaign was conducted in the straight tunnel of Shanghai metro Line 7 (tunnel from Shanghai University station to Qihua Road station). The considered frequencies are at the 3.5 GHz and 5.6 GHz bands. The cross-section of the tunnel is arched, with a radius of 2.78 m and a bottom of 3.4 m, as shown in Figure 6a.

In the measurement, the transmitting and receiving antennas are located on the railcar in the center of the tunnel. The height of both antennas is 2.65 m, and the positions of the transmitting and receiving antennas are shown in Figure 6b. When the transceiver distance is from 20 m to 350 m, the sampling interval is 10 m, and the interval from 350 m to 650 m is 50 m. There is a total of 40 measuring points.

The configuration of the measurement system is shown in Figure 7. It mainly comprises an Agilent E8257D signal source and Ceyear 4024G spectrum analyzer. Both transmitting (T_x_) and receiving (R_x_) antennas are ultra-wideband omnidirectional antennas, and the rubidium clock is used to ensure the clock synchronization. The input power is set as 10 dBm. As the railcar moves, the receiver gradually moves away from the transmitter, and the spectrum analyzer reads and records the received power of the signal at each test position. The antenna is high enough to ignore the influence of the testers.

### 3.2. RT Parameters Calibration

In Figure 8, we reconstructed the 3D model of the actual metro tunnel and then imported it into Wireless Insite (RT-based channel simulation software) [40]. During the RT simulation, the antenna type, frequency, and locations are consistent with the actual measured scenario.

To ensure that the simulation is consistent with the actual measurement, the parameters of material property in the 3D tunnel model need to be set properly, as shown in Table 2. The permittivity and conductivity of the materials employ the values recommended by ITU-R P.2040 [41]. The roughness of concrete is set as 0.075, and the surface of the metal is smooth enough that its roughness could be ignored in RT simulation [42]. In order to generate channel characteristics close to the actual measurement, the parameters in the RT simulation are calibrated according to the path loss of the field measurement [43,44]. After several simulation experiments, the number of reflections is set as 23, and the fine-tuned spacing of rays is set as 0.12°.

As shown in Figure 9, we use the classic floating intercept path-loss model [45] to fit the RT simulation and the measurement. The equation is described as follows:(24)$$\mathrm{PL}(d)=\beta +10\alpha log\;(\frac{d}{d_{0}})+X_{\sigma }\left(\mathrm{dB}\right)$$
where *α* is the slope of the PL model, and *d*_0_ is the reference distance, which is 1 m in this study. *β* is the model intercept which represents the path loss when the transceiver distance is the reference distance, and *X_σ_* is a Gaussian random variable with a standard deviation of *σ*.

As shown in Table 3, it can be seen that the PL model for the actual measurement campaign and the RT simulation match very well, and the difference in parameters (*α*, *β*, and *X_σ_*) between the actual measurement campaign and RT simulation is quite small. The comparison results confirm that the RT simulation parameters are very close to the values in the actual environment, indicating that the corrected material parameters can be used to simulate in different subway tunnel environments [32].

## 4. Determination of ELC by RT Simulation

To simulate the cascaded tunnel, a three-dimensional model composed of a straight tunnel with a length of 600 m and a curved tunnel with a length of 400 m is established. As shown in Figure 10a, T_x_ is in the straight tunnel, and the distance of T_x_ from the curved tunnel is *d*. R_x_ is placed after every 10 m in the curvature (40 positions in total). To extract the EL generated by the curve, a straight tunnel model with the same dimensions is established, as shown in Figure 10b, and the antenna positions are identical to those used in the cascaded tunnel. The cross-section of both tunnels and the height of the antennas are the same as shown in Figure 6a.

The above two models and antenna configurations are imported into Wireless Insite for channel simulations. The simulation parameters are shown in Table 4. According to (2)–(5), the arched tunnel is equivalent to a rectangular tunnel with *w* = 4.73 m and *h* = 4.23 m. Then, the BP of each frequency band is calculated by using (1), and three different *d* are selected for each frequency band using (23), as shown in Table 3.

From the results of RT simulation, it can be found that the curvature of the cascaded tunnel will produce obvious EL compared with the *PL_straight_* in the straight tunnel when *f* = 3.5 GHz, *d* = 300 m, and *R* = 300 m, as shown in Figure 11a. This phenomenon is due to the transition of the waveguide mechanism when there is a curvature in tunnels. As the direct path is blocked, the enhanced waveguiding mechanism would be taken over by degraded waveguiding, resulting in greater path loss. Taking *PL_straight_* as the reference value, the EL can be extracted and fitted with the direct proportional function in Figure 11b.

In Figure 11b, EL is approximately proportional to the propagation distance in the curvature, which is consistent with Equation (12). The root mean square error (RMSE) between EL and the fitting line is 1.39 dB, and the slope of the fitting line is the ELC. The ELC and the average RMSE for different radii of curvature (R) are calculated in Table 5.

For the 3.5 GHz and 5.6 GHz bands, the average RMSE is between 1.35 dB and 2.17 dB, which justifies that EL can be well fitted into a linear function. In addition, the value of *d* has little effect on the ELC when *d* ≥ BP. As R decreases, the ELC evidently increases, which indicates that smaller radius of curvature leads to larger EL at the same distance. This is because more radio waves will be blocked, creating NLOS environments with propagation by reflections only. On the other hand, the ELC decreases around 3 dB/100 m when R increases from 300 m to 900 m, and the ELC decreases around 1 dB/100 m when R increases from 900 m to 1500 m. From the results, the relationship between ELC and R follows an approximate power function, which is consistent with Equation (19).

In Figure 12, the ELCs at 5.6 GHz are slightly higher than 3.5 GHz under the same radius of curvature, which means the higher frequencies are more sensitive to the curvature. This finding implies that EL caused by curvature would nullify a lower propagation attenuation rate at higher frequencies, resulting in the increase in total attenuation at higher frequencies in cascaded tunnels. Hence, the influence of frequency and the curvature should be considered together when predicting path loss in cascaded tunnels.

In addition, the fitting parameters in Equation (19) and the root mean square error (RMSE) between the value of ELC and fitted values are shown in Table 6. The fitting function can be used as an empirical formula to calculate ELC at 3.5 GHz and 5.6 GHz in the cascaded tunnel.

## 5. Verification and Comparison

To verify the accuracy of the proposed model, 3.5 GHz, and 5.6 GHz bands are simulated in cascaded tunnels with different curvatures (*R* = 500 m and 1000 m). The ELC for different scenarios is calculated through the empirical formula, and the results are given in Table 7. To ensure that the curvatures are entirely in the far-field region, the propagation distances *d* at 3.5 GHz and 5.6 GHz in the straight tunnel are set as 400 m and 500 m, respectively.

The path loss of a typical RT model relying on high-precision 3D tunnel models [32] at 3.5 GHz and 5.6 GHz for different radii of curvature is also given for comparison. As shown in Figure 13 and Figure 14, the proposed model matches well with the results of the computationally intensive RT model [32]. For the 3.5 GHz band, the RMSE in the cascaded tunnels with R = 500 m and 1000 m are 2.19 dB and 1.68 dB, respectively. Meanwhile, for the 5.6 GHz band, the RMSEs are only 1.76 dB and 1.71 dB, showing a good agreement with the results of the RT model.

Since a prerequisite of the RT model is the 3D tunnel model constructed to describe the specified tunnel environments, the complexity per channel instance of the RT model is *O*(*NT^N^*^+1^), where *N* is the number of reflections and *T* is the total number of the basic surface units in the 3D model [46]. Considering typical values of *N* and *T* in the order of 2–6 and 100–10,000, respectively, the RT model requires excessive memory and CPU resources to run large-scale numerical electromagnetic simulations.

Compared with the RT model, the simplified model proposed in this paper can predict path loss in cascaded tunnels with comparable accuracy but significantly lower complexity, which eases the demands on computational resources and facilitates propagation modeling for cascaded tunnels.

## 6. Conclusions

Combining the waveguide mode with the method of shooting and bouncing ray, this paper studied the extra loss in the far-field region, which can be modeled as a linear function of the propagation distances in the curvature section of the tunnel, and an expression for ELC was proposed. Then, channel measurements at 3.5 GHz and 5.6 GHz bands were conducted in a real subway tunnel environment and the material parameters of RT simulation were calibrated according to path loss observed in the measurement campaign. An empirical formula between the extra loss coefficient and the radius of the curvature was determined for 3.5 GHz and 5.6 GHz bands.

The model proposed in this paper achieved comparable accuracy with existing RT models, with significant savings in execution time and memory. It is applicable to a straight tunnel followed by a curved tunnel with a given radius of curvature. A more general path-loss model for longer cascades with multiple straight and curved tunnels is an interesting topic for future research and deserves further investigation. Moreover, the impacts of polarization, antenna position and tunnel dimension on the extra loss coefficient need to be further studied, which will be helpful for establishing a more unified and adaptable path-loss model in cascaded tunnel environments.

## Figures and Tables

**Figure 1 sensors-22-04524-f001:**
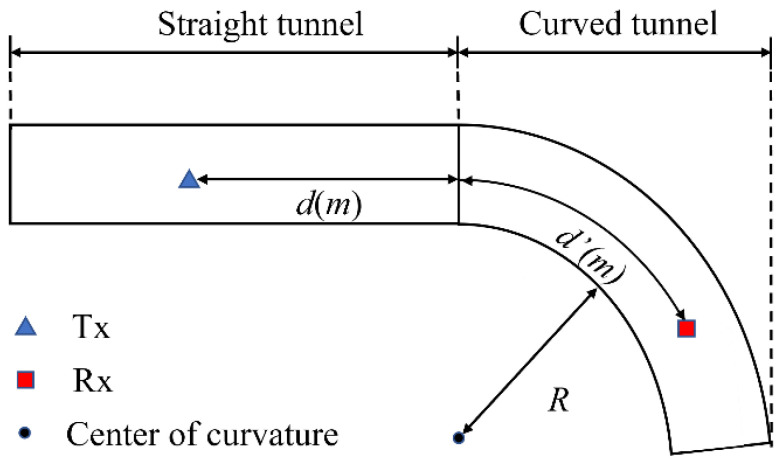
Cascaded tunnel and antenna locations.

**Figure 2 sensors-22-04524-f002:**
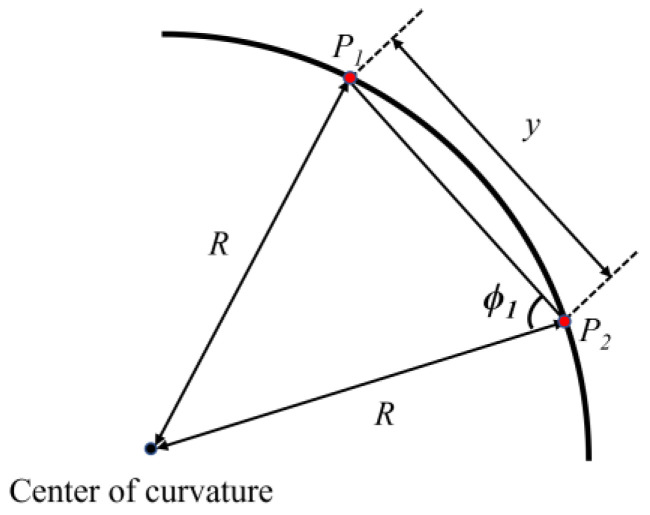
Schematic diagram of reflection in cascaded tunnel.

**Figure 3 sensors-22-04524-f003:**
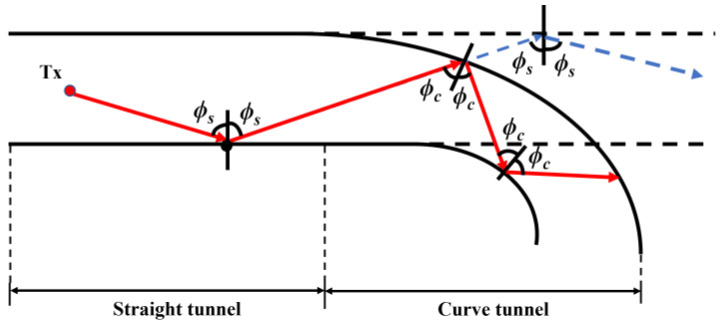
Schematic diagram of reflection in cascaded tunnel.

**Figure 4 sensors-22-04524-f004:**
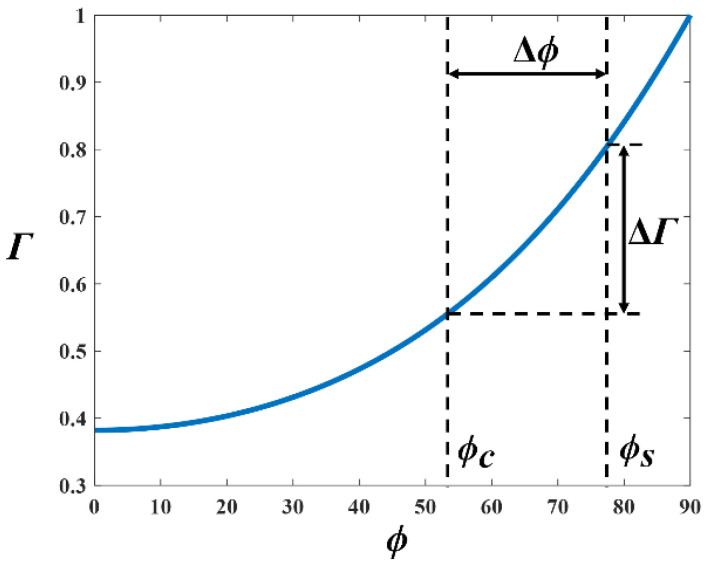
The relationship between incident angle and reflection coefficient.

**Figure 5 sensors-22-04524-f005:**
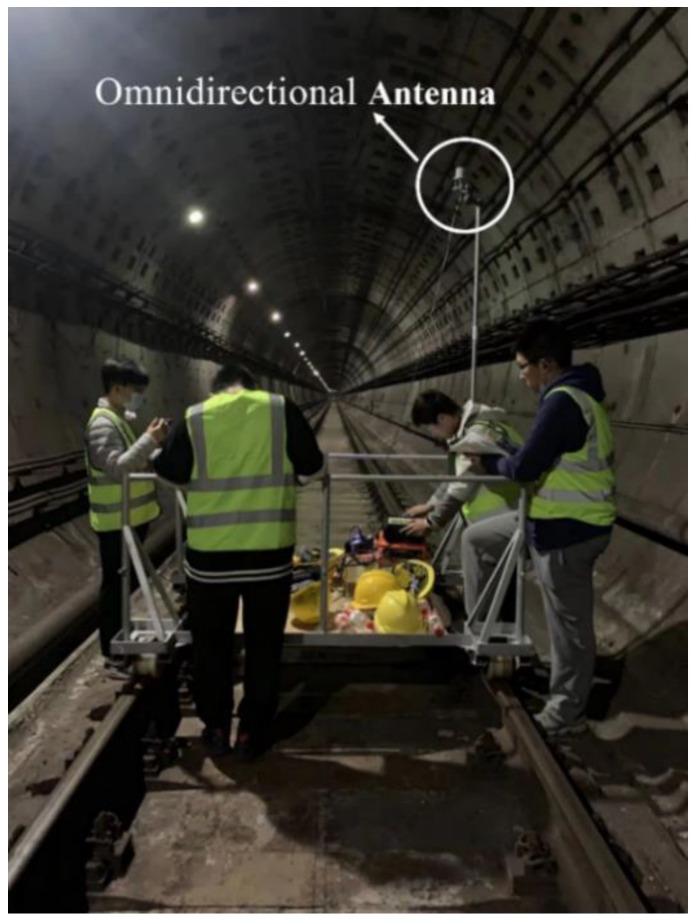
The measurement environment.

**Figure 6 sensors-22-04524-f006:**
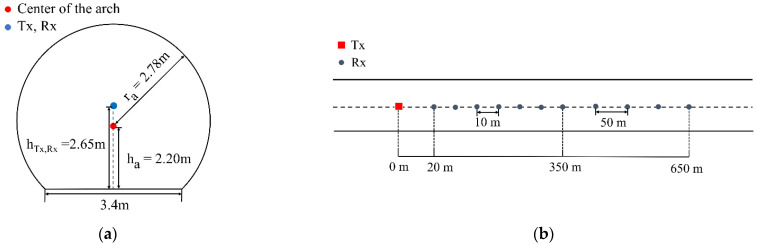
Transmitting and receiving antennas setting (**a**) The cross-section of the measurement tunnel; (**b**) the positions of T_x_ and R_x_.

**Figure 7 sensors-22-04524-f007:**
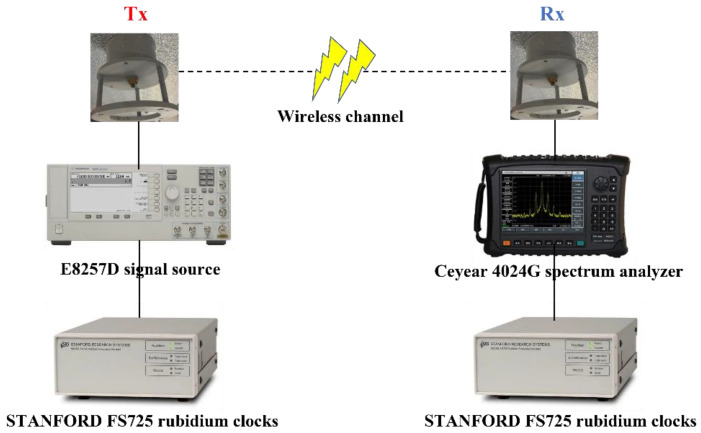
The configuration of the measurement system.

**Figure 8 sensors-22-04524-f008:**
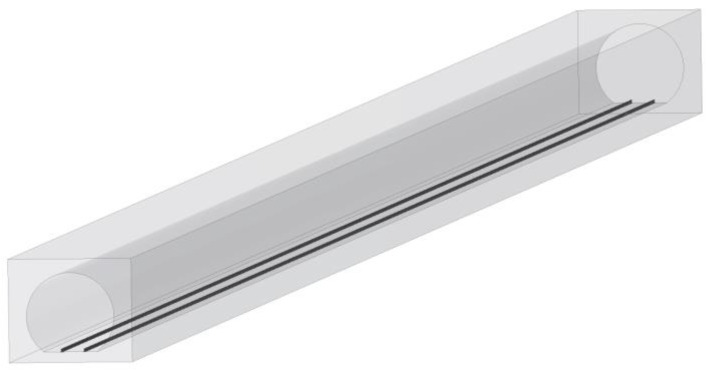
The 3D model of the straight tunnel.

**Figure 9 sensors-22-04524-f009:**
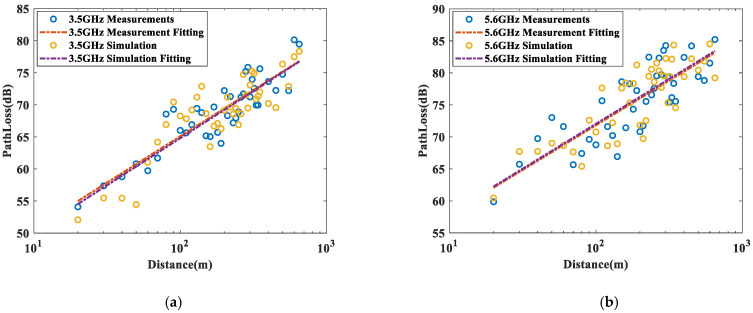
Comparison of path-loss results between measurement and RT simulation (**a**) 3.5 GHz (**b**) 5.6 GHz.

**Figure 10 sensors-22-04524-f010:**
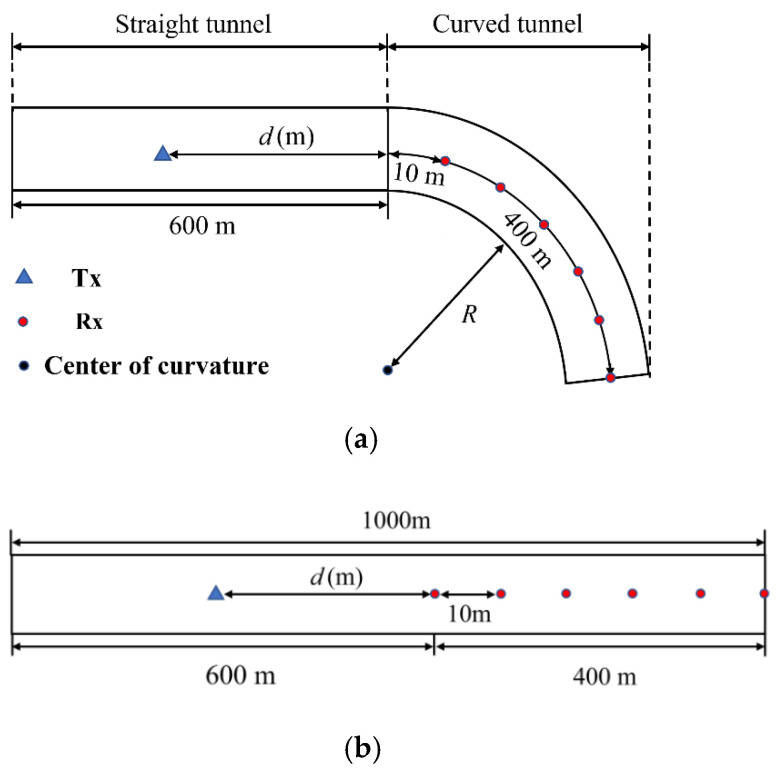
The model and antennas setting (**a**) Cascaded tunnel; (**b**) straight tunnel.

**Figure 11 sensors-22-04524-f011:**
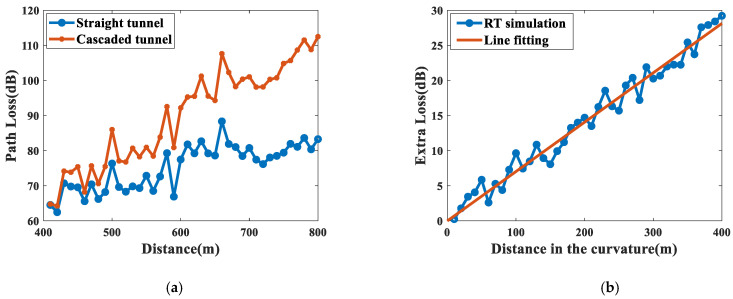
(**a**) Comparison of path loss between straight tunnel and cascaded tunnel; (**b**) EL fitted by direct proportional function.

**Figure 12 sensors-22-04524-f012:**
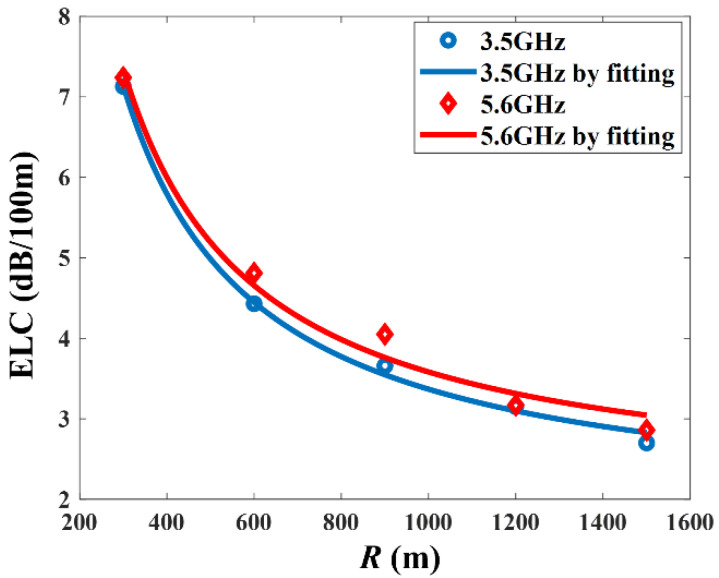
The relationship between ELC and *R* at 3.5 GHz and 5.6 GHz.

**Figure 13 sensors-22-04524-f013:**
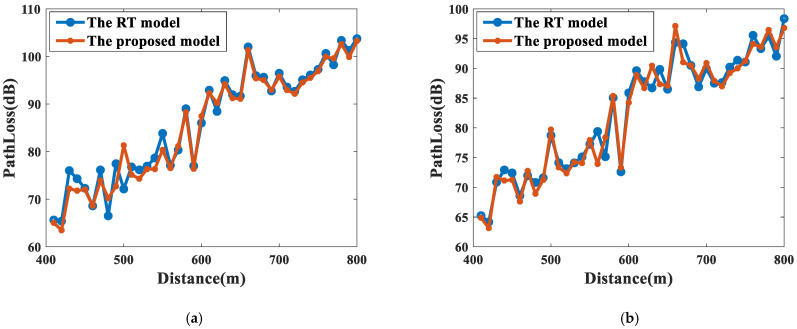
The comparison between the RT model in [32] and the proposed model at 3.5 GHz. (**a**) R = 500 m; (**b**) R = 1000 m.

**Figure 14 sensors-22-04524-f014:**
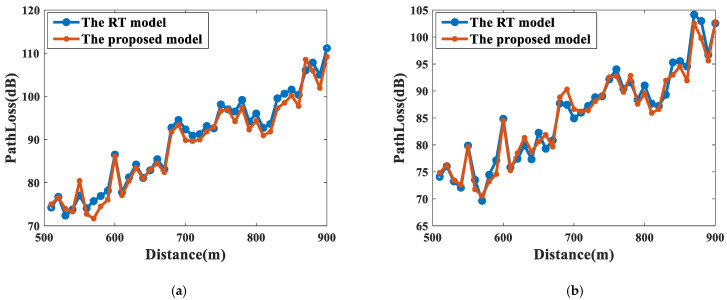
The comparison between the RT model in [32] and the proposed model at 5.6 GHz. (**a**) R = 500 m; (**b**) R = 1000 m.

**Table 1 sensors-22-04524-t001:** State-of-the-art related works on path-loss models in tunnel environments.

Type	Methods	Models	Pros and Cons	Ref.
Straight tunnel	Fit measurement results using regression method	FI model	Low complexity Insufficient accuracy	[21]
	Superpose multiple modes in both near and far region	Multimode model	High accuracy Limited applicability	[22]
	Calculate Per-ray cone angle	Improved RT model	High accuracy Low computational efficiency	[23]
	Extract rectangular waveguide model using VPE	Mixed model based on waveguide and VPE	Reduced complexity Limited Validity	[24,25]
Curved tunnel	Introduce a break point distance into the CI model	Improved CI model	High accuracy Less stability	[26]
	Divide propagation region into LOS and NLOS	Two-slope model	Realistic scenario Large deviation	[27]
	Define the break point between two waveguiding effects	Improved FI model with break point	High accuracy Calculations of break point required	[28]
	Estimate the main effects of the curvature on multimode	Mixed model based on waveguide and RT	Low complexity Insufficient accuracy	[29]
	Combine RT method with neural network	Improved RT model	High applicability High complexity	[30]
Cascaded tunnel	Fit measurement results using regression method	CI model	Low complexity Insufficient accuracy	[31]
	Reconstruct a high-precision 3D model of measurement tunnel	RT model	High accuracy High-precision 3D model required	[32]
	Divide space into segments to solve stability constraint	Improved FDTD model	High accuracy high complexity	[33]

**Table 2 sensors-22-04524-t002:** Material parameters after calibration.

	Material	Roughness (m)	Conductivity (S/m)	Permittivity
Tunnel Wall	Concrete	0.075	0.09 (3.5 GHz)/0.15 (5.6 GHz)	5.31
Rail	Metal	-	10^7^	5.31

**Table 3 sensors-22-04524-t003:** Path-loss model parameters at 3.5 GHz and 5.6 GHz bands.

	*α*	*β*	*X_σ_*
3.5 GHz Measurement	1.444	36.217	2.506
3.5 GHz RT simulation	1.461	35.437	2.970
5.6 GHz Measurement	1.394	43.938	3.393
5.6 GHz RT simulation	1.405	44.021	3.252

**Table 4 sensors-22-04524-t004:** BP and d at 3.5 GHz and 5.6 GHz bands.

*f* (GHz)	*d*_1_ (m)	*d*_2_ (m)	*d*_3_ (m)	BP (m)
3.5	300	350	400	261
5.6	450	500	550	417

**Table 5 sensors-22-04524-t005:** ELC and average RMSE for different radii of curvatures.

*f* (GHz)	*R* (m)	ELC (dB/100 m)			Average RMSE (dB)
		*d*_1_ (m)	*d*_2_ (m)	*d*_3_ (m)	
3.5	300	7.18	7.03	7.17	1.66
3.5	600	4.35	4.28	4.66	1.69
3.5	900	3.45	3.56	3.96	1.35
3.5	1200	3.25	3.16	3.05	1.89
3.5	1500	2.82	2.62	2.65	1.95
5.6	300	7.22	7.35	7.15	1.81
5.6	600	4.56	5.32	4.55	2.17
5.6	900	4.12	4.15	3.89	1.72
5.6	1200	3.09	3.16	3.26	1.58
5.6	1500	2.82	2.99	2.77	1.55

**Table 6 sensors-22-04524-t006:** Path-loss model parameters at 3.5 GHz and 5.6 GHz bands.

*f* (GHz)	*A*	*b*	RMSE (dB)
3.5	1.75	1618	0.104
5.6	1.97	1612	0.230

**Table 7 sensors-22-04524-t007:** Calculation of ELC using empirical formula.

*f* (GHz)	*R* (m)	ELC (dB/100 m)
3.5	500	5.00
3.5	1000	3.38
5.6	500	5.20
5.6	1000	3.58

## Data Availability

Not applicable.

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
