# Peer review of "Path Loss Model for 3.5 GHz and 5.6 GHz Bands in Cascaded Tunnel Environments"

_sensors, 2022, doi:10.3390/s22124524_

Round 1

Reviewer 1 Report

In this paper, a simulation study is performed to investigate the path loss model for cascaded tunnels at 3.5 GHz and 5.6 GHz bands. Overall, I believe that the research has some contributions. I suggest that this paper be revised to improve its quality.
1.  The introduction section needs to be improved by providing a table to summarize the state of art schemes, with their techniques, advantages and limitations.
2.  Please enhance reasonable analyses and discussions in section 4 and section 5.
3. The article should present the limitations of the work and directions for future research in the conclusion, which means conclusion section should be further enhanced.
4. Some important references are missing. Most references are very old. A more comprehensive literature review is required to present the state of the art.

Reviewer 2 Report

1- The problem statements in the introduction section are not clear.

2- The main contribution also not clear.

3- The results and discussion require needs more explanation to understand the

finding more clearly.

4- The analysis is also poor. Improve it by compare it with recent techniques and evaluate the throughput of the method.

5- The discussion regarding the performance enhancement of the proposed method with existing techniques should be included.

6- The authors need to refer some latest references. References are not much recent.

7- Literature survey should include latest references for drawing a comparative analysis of the proposed approach.

8- Improve the technical writing with more explanation of the proposed work with proper justification.

9- The earlier work on the scope of the paper may be given in tabular format to draw any comparison.

10- The results are not enough to show the contributions of this paper, more results and comparisons need to provide.

11- The authors should add a section "Related Works" to demonstrate adequacy.

12- The authors did not mention with which references the comparison done in figures in simulation results section.

Reviewer 3 Report

This work proposes a joint path loss model for cascaded subway tunnels, composed of straight and curved segments. Existing modeling methods (waveguide and SBR) are combined and ray-tracing simulation is used for parameter fitting and validation against real-world measurements at 3.5 and 5.6 GHz.

The study appears technically sound and relevant to the Journal focus. However, there are several recurring grammatical flaws in the text (missing articles, inadequate use of singular/plural, wrong wording) that should be corrected before accepting the paper. Please find detailed comments marked in yellow in the attached PDF.

In the discussion section, the authors could give some indication on the applicability of their joint model to arbitrary tunnel cascades. In the study, a cascade of a single straight and a single curved tunnel was investigated. Can the method be applied to longer cascades without limitations?

Round 2

Reviewer 1 Report

The paper has been properly revised according to the reviewers’ comments. It can be accepted now.  

Reviewer 2 Report

they answered all my concerns